# Whole-Lesion Apparent Diffusion Coefficient Histogram Analysis: Significance for Discriminating Lung Cancer from Pulmonary Abscess and Mycobacterial Infection

**DOI:** 10.3390/cancers13112720

**Published:** 2021-05-31

**Authors:** Katsuo Usuda, Shun Iwai, Aika Yamagata, Yoshihito Iijima, Nozomu Motono, Munetaka Matoba, Mariko Doai, Keiya Hirata, Hidetaka Uramoto

**Affiliations:** 1Department of Thoracic Surgery, Kanazawa Medical University, Ishikawa 920-0293, Japan; mhg1214@kanazawa-med.ac.jp (S.I.); aicarby@kanazawa-med.ac.jp (A.Y.); y-iijima@kanazawa-med.ac.jp (Y.I.); motono@kanazawa-med.ac.jp (N.M.); hidetaka@kanazawa-med.ac.jp (H.U.); 2Department of Radiology, Kanazawa Medical University, Ishikawa 920-0293, Japan; m-matoba@kanazawa-med.ac.jp (M.M.); doaimari@kanazawa-med.ac.jp (M.D.); 3MRI Center, Kanazawa Medical University Hospital, Ishikawa 920-0293, Japan; keiya@kanazawa-med.ac.jp

**Keywords:** diffusion-weighted magnetic resonance imaging (DWI), magnetic resonance imaging (MRI), lung cancer, pulmonary abscess and mycobacterial infection (PAMI), apparent diffusion coefficient (ADC), ADC histogram

## Abstract

**Simple Summary:**

Diffusion-weighted magnetic resonance imaging (DWI) can differentiate malignant from benign pulmonary nodules and masses. However, it is difficult to differentiate pulmonary abscesses and mycobacterium infections (PAMIs) from lung cancers because PAMIs show restricted diffusion in DWI. The purpose of this study was to establish the role of ADC histogram for differentiating lung cancer from PAMI. There were 41 lung cancers and 19 PAMIs. Parameters more than 60% of AUC were ADC, maximal ADC, mean ADC, median ADC, most frequency ADC, kurtosis of ADC, and volume of lesion. There were significant differences between lung cancer and PAMI in ADC, mean ADC, median ADC, and most frequency ADC. ADC histogram has the potential to be a valuable tool to differentiate PAMI from lung cancer.

**Abstract:**

Diffusion-weighted magnetic resonance imaging (DWI) can differentiate malignant from benign pulmonary nodules. However, it is difficult to differentiate pulmonary abscesses and mycobacterial infections (PAMIs) from lung cancers because PAMIs show restricted diffusion in DWI. The study purpose is to establish the role of ADC histogram for differentiating lung cancer from PAMI. There were 41 lung cancers (25 adenocarcinomas, 16 squamous cell carcinomas), and 19 PAMIs (9 pulmonary abscesses, 10 mycobacterial infections). Parameters more than 60% of the area under the ROC curve (AUC) were ADC, maximal ADC, mean ADC, median ADC, most frequency ADC, kurtosis of ADC, and volume of lesion. There were significant differences between lung cancer and PAMI in ADC, mean ADC, median ADC, and most frequency ADC. The ADC (1.19 ± 0.29 × 10^−3^ mm^2^/s) of lung cancer obtained from a single slice was significantly lower than that (1.44 ± 0.54) of PAMI (*p* = 0.0262). In contrast, mean, median, or most frequency ADC of lung cancer which was obtained in the ADC histogram was significantly higher than the value of each parameter of PAMI. ADC histogram could discriminate PAMIs from lung cancers by showing that AUCs of several parameters were more than 60%, and that several parameters of ADC of PAMI were significantly lower than those of lung cancer. ADC histogram has the potential to be a valuable tool to differentiate PAMI from lung cancer.

## 1. Introduction

In pulmonary nodules and masses (PNMs) lung cancer is one of the leading causes of cancer-related deaths and its correct diagnosis is essential for all patients. For the imaging method of PNMs, 18-fluoro-2-deoxy-glucose positron emission tomography/computed tomography (FDG-PET/CT) has been widely performed. Its maximum standardized uptake value (SUVmax) shows glucose uptake and indicates how aggressive the lesion is. FDG-PET/CT is useful for discriminating benign from malignant pulmonary nodules [1]. The FDG-PET/CT weak points are expressing false-positive results for inflammatory nodules [2], and false-negative results for well-differentiated pulmonary adenocarcinoma [3] and small volumes of metabolically active tumors [4].

In magnetic resonance imaging (MRI), diffusion-weighted magnetic resonance imaging (DWI) has been developed for detecting the restricted diffusion of water molecules. The random motion of water molecules in biological tissues is presented in DWI [5]. DWI can discriminate malignant from benign pulmonary lesions in a meta-analysis report [6]. In DWI, discrimination between benign and malignant lesions was possible in the lung [6,7], in the thorax [8], in the prostate [9], in the breast [10], and in the liver [11]. In actuality, it is difficult to discriminate pulmonary abscesses and mycobacterial infections (PAMIs) from lung cancers because PAMIs show restricted diffusion in DWI.

The ADC value is calculated using one slice of the ADC map which limits the accuracy of the evaluation due to the fact it does not catch the whole lesion. Recently, ADC histograms which analyze the whole tumor were reported to be useful for malignancy evaluation [12,13]. We have already reported that ADC histogram analyses on the basis of the entire tumor volume was able to stratify non-small cell lung cancer’s tumor grade, lymphovascular invasion, and pleural invasion [14].

The purpose of this study is to establish the role of ADC histograms for differentiating lung cancer from PAMI. If the lesion is a suspected PAMI, determined by the ADC histogram, we can avoid an unnecessary operation for the lesion.

## 2. Materials and Methods

### 2.1. Eligibility

The study protocol for examining DWI in patients with lung cancers and PAMIs was approved by the ethical committee of Kanazawa Medical University (the approval number: No. I302). Written informed consents for MRIs were obtained from each patient after discussing the risks and benefits of the examinations. All methods were carried out in accordance with the relevant guidelines and regulations of the Declaration of Helsinki.

### 2.2. Patients

This is a retrospective study. Patients included in the study had PNMs with a maximum size of 15 cm or less, and which had no definitive calcification. Patients who had metal or pacemakers in their body or tattoos on the skin were excluded because of contraindication in MRI examinations. Pure ground-glass-opacity (GGO)-type lung cancers were excluded in this study because all pure GGO-type lung cancers were negative in DWI.

PNMs, which were diagnosed as lung cancer or PAMI by resection or bacterial culture from May 2009 to April 2018, were included in the study and analyzed in ADC histograms of all the pixels of the whole lesions. Five patients were excluded from the analysis; accurate diagnosis was not obtained in 2 patients, and the measurements of ADC in the other 3 patients were not done because the small lesions could not be detected in the ADC histogram. Sixty patients with a primary lung cancer of adenocarcinoma, squamous cell carcinoma, or PAMI were enrolled in this study (Table 1). None of the patients had received prior treatment. Thirty-nine patients were male and 21 were female. Their mean age was 70 years old (range 47 to 84). There were 41 lung cancers and 19 PAMIs. The diagnosis was made by resection pathologically in 41 lung cancers and 14 PAMIs, and by a bacterial culture in 5 PAMIs. For 41 lung cancers there were 25 adenocarcinomas and 16 squamous cell carcinomas. For 19 PAMIs, there were 9 pulmonary abscesses and 10 mycobacterial infections (tuberculosis 3, nontuberculous mycobacteria 7).

### 2.3. MR Imaging

All MR images were produced with a 1.5 T superconducting magnetic scanner (Magnetom Avanto; Siemens, Erlangen, Germany) with two anterior six-channel body phased-array coils and two posterior spinal clusters (six-channels each). The conventional MR images consisted of a coronal T1-weighted spin-echo sequence and coronal and axial T2-weighted fast spin-echo sequences. DWIs using a single-shot echo-planar method were applied with slice thickness of 6 mm under SPAIR (spectral attenuated inversion recovery) with respiratory triggered scan with the following parameter: TR/TE/flip angle, 3000-4500/65/90; diffusion gradient encoding in three orthogonal directions; b value = 0 and 800 s/mm^2^; field of view, 350 mm; matrix size, 128 × 128. A MR examination usually takes about 30 min. After image reconstruction, region of interest (ROI) for the ADC was set up. ADC values were obtained by drawing round, elliptical, or free-hand regions of interest (ROIs) on lesions which were detected visually on the ADC map with reference to T2-weighted or CT image (the original source of method description was cited from Usuda K, et al. [15]). Areas with necrosis were excluded from the ADC measurement. A radiologist with 25 years of MRI experience who was unaware of the patients’ clinical data performed these measurements. The procedures were performed three times and the minimum ADC value was obtained. A newly developed medical imaging software, BD score (PixSpace, Fukuoka, Japan) was used for the analysis of ADC histograms. On BD score views and ADC histograms of the lesions, first ADC area (0.1–0.5 × 10^−3^ mm^2^/s) were presented in red, second ADC area (0.5–1.0 × 10^−3^ mm^2^/s) in yellow, and third ADC area (1.0–2.0 × 10^−3^ mm^2^/s) in green (Figure 1 and Figure 2). Using the BD score a pulmonologist with 30 years of experience took the ADC histograms which were a visual representation of all the pixels of the entire lesions. In this study, ADC (mean apparent diffusion coefficient value obtained from one region of interest in single slice) and parameters of ADC histograms (minimum ADC, maximum ADC, mean ADC, median ADC, standard deviation of ADC, most frequency ADC, kurtosis of ADC, skewness of ADC and volume of lesion) were analyzed.

### 2.4. Statistical Analysis

The data is expressed as the mean ± standard deviation. A two-tailed Student t-test was applied for comparison of ADC values in several prognostic factors. A two-tailed Student t-test was performed for comparison of several values of two groups and ANOVA was performed for comparison of several values of three or more groups in several factors. Using GraphPad Prism (Version 5.02, GraphPad Software, Inc., La Jolla, CA, USA) receiver operating characteristic (ROC) curves were obtained and optimal cutoff values (OCVs) of the ADC and parameters of ADC histogram in terms of discrimination of lung cancers from PAMI were determined. The statistical analyses were carried out using the computer software program StatView for Windows (Version 5.0; SAS Institute Inc. Cary, NC, USA). A *p*-value of < 0.05 was considered statistically significant.

## 3. Results

Chest CTs, DWIs, ADCmaps, BD score views, and ADC histograms of the lesions were presented (Figure 1 and Figure 2).

Patient No.1 had an adenocarcinoma with ADC (0.96 × 10^−3^ mm^2^/s) obtained on single slice (Figure 1). Patient No.2 had a pulmonary abscess with ADC (1.14 × 10^−3^ mm^2^/s) obtained on single slice (Figure 2).

Using the receiver operating characteristic (ROC) curve the performance of ADC and ADC histogram parameters for differential diagnosis for lung cancer and PAMI was shown in Table 2. Parameters more than 60% of area under the ROC curve (AUC) were ADC, maximal ADC, mean ADC, median ADC, most frequency ADC, kurtosis of ADC, and volume of lesion (Figure 3). The AUC performance of most of the variables was more than 60% but did not reach 70%. The one outlier was the volume of lesions in our study that did reach 70%. Six of the nine variables had AUC scores higher than 60%. AUC showed better sensitivity, specificity and positive predictive value (PPV) more than 60%. Volume of lesion (AUC 70.9%) showed sensitivity 75.6%, specificity 68.4%, PPV 83.8% and negative predictive value (NPV) 56.5%. Median ADC (AUC 69.3%) showed sensitivity 75.6%, specificity 63.2%, PPV 81.6% and NPV 54.5%. Mean ADC (AUC 68.6%) showed sensitivity 80.5%, specificity 63.2%, PPV 82.5% and NPV 60.0%. Most frequency ADC (AUC 66.5%) showed sensitivity 73.2%, specificity 63.2%, PPV 81.1% and NPV 52.2%. Kurtosis of ADC (AUC 66.0%) showed sensitivity 75.6%, specificity 68.4%, PPV 83.8% and NPV 56.5%. Maximal ADC (AUC 65%) showed sensitivity 68.3%, specificity 57.9%, PPV 77.8% and NPV 45.8%.

Comparisons of ADC or parameters of ADC [minimum ADC, maximum ADC, mean ADC, median ADC, standard deviation (A.D.) of ADC, most frequency ADC, kurtosis of ADC, skewness of ADC, and volume of lesion between lung cancer and PAMI were presented in Table 3. In these parameters, there were significant differences between lung cancer and PAMI in ADC, mean ADC, median ADC, most frequency ADC (Figure 4). ADC (1.19 ± 0.29 × 10^−3^ mm^2^/s) of lung cancer was significantly lower that (1.44 ± 0.54) of PAMI (*p* = 0.0262). On the contrast mean ADC (1.21 ± 0.21 × 10^−3^ mm^2^/s) of lung cancer was significantly higher than that (1.05 ± 0.30 × 10^−3^ mm^2^/s) of PAMI (*p* = 0.0265). Median ADC (1.18 ± 0.22) of lung cancer was significantly higher than that (1.03 ± 0.31) of PAMI (*p* = 0.0301). Most frequency ADC (1.00 ± 0.27 × 10^−3^ mm^2^/s) of lung cancer was significantly higher that (0.80 ± 0.41 × 10^−3^ mm^2^/s) of PAMI (*p* = 0.0254).

In ADC histograms of lung cancer and PAMI, these parameter of mean ADC, median ADC, most frequency ADC of lung cancer were revealed to be significantly higher than each parameter of PAMI, which was an opposite of results of ADC (mean apparent diffusion coefficient value obtained from one region of interest in single slice).

## 4. Discussion

Three meta-analyses of DWI were reported for the discriminant diagnosis of malignancy and benignity for PNMs [6,16,17] and they concluded that DWI could discriminate malignancy from benignity for PNMs. However, it is difficult to discriminate PAMIs from lung cancers because PAMIs showed strong restricted diffusion in DWI. In general, the ADC of malignant tumors was usually significantly lower than that of normal tissues or benign lesions [18], but in fact the ADC of a PAMI is lower than or similar to that of lung cancer. Through the analysis of ADC histograms between lung cancer and PAMI, these parameters of mean ADC, median ADC, and most frequency ADC of lung cancer were revealed to be significantly higher than those of PAMI, which was an opposite result to ADC (mean apparent diffusion coefficient value obtained from one region of interest in single slice). Its opposite results were considered as follows: Conventionally, necrotic areas were excluded from the ADC measurements for lung cancer or other lesions because necrotic areas were inadequate tissue for analysis [15]. The ADC values of necrotic areas are also important for PAMI. Most PAMI cases in our study have necrotic areas and we concluded that we should measure the ADC values both in necrotic areas and non-necrotic areas for precise discrimination for ADC of lung cancer and PAMI. Using a sample area without necrosis is connected to higher ADC values of lung cancer and PAMI. In contrast, ADC histograms of lung cancer and PAMI analyzed the whole lesion containing necrosis. Resulting parameters of ADC histograms showed a decrease in ADC values. This study presents that ADC histograms have the ability to look at whole lesions in their entirety using automated calculations and have the potential to be a valuable tool in differentiating lung cancer from PAMI. Measuring whole lesions and automating measurements is not only beneficial in differentiating lung cancer from PAMI but for also standardizing this data when taking ADC measurements.

The AUC performance of most of the variables was more than 60% but did not reach 70%. The one outlier was the volume of lesions in our study that did reach 70%. Six of the nine variables had AUC scores higher than 60%. AUC showed better sensitivity, specificity and PPV more than 60%. While a 70% AUC performance is stronger data, an AUC performance of more than 60% is still valuable data. In the future by innovation of a software for analyzing ADC histogram in MRI examinations, ADC histograms and the ROC curve will be useful for the discrimination of indistinguishable lesions.

Some pathologic processes such as pulmonary tuberculosis, nontuberculous mycobacteria, pulmonary abscesses, sarcoidosis, chronic pneumonia, scars, and other inflammatory or infectious conditions behave like malignant lesions by exhibiting diffusion restriction [19,20,21]. ADC values of abscesses are low, and it was mentioned that median ADC value (0.877 × 10^−3^ mm^2^/s) of abscesses was significantly lower than that (2.118 × 10^−3^ mm^2^/s) of phlegmon (*p* < 0.001), and that (3.008 × 10^−3^ mm^2^/s) of edema (*p* < 0.01) [22]. There were several reasons for the restricted diffusion of PAMI. Abscesses and thrombi impede the diffusivity of water molecules because they possess a hyperviscous nature [23,24]. Low ADC values of necrosis were related with the organized abscess environment containing microorganisms, macromolecules, and intact inflammatory cells [25]. The heavily impeded mobility of pus can be explained by its high cellularity and viscosity and shows low ADC values [26]. The possible causes were due necrotic debris, viscosity, a combination of cells, and macromolecules present in the pus [24,26]. Most brain abscesses have low ADC values, whereas non-abscess (tumor) groups have high ADC values [27]. The ADC (1.11 × 10^−3^ mm^2^/s) of lung cancer with necrosis was significantly lower than that (1.32 × 10^−3^ mm^2^/s) of lung cancer without necrosis (*p* = 0.0001) [28].

In two papers, diagnostic efficacy of DWI was compared with that of FDG-PET/CT for PNMs [7,21]. One paper showed that the sensitivity and the accuracy of DWI were significantly higher than that of FDG-PET/CT [7], and the other showed that the sensitivity of DWI was significantly higher than those of FDG-PET/CT [21]. DWI possess higher potential than FDG-PET/CT in assessing PNMs.

ADC histogram would be a valuable tool to differentiate between morphologically indistinguishable mass lesions and have an advantage in differential diagnosis in a viewpoint of information of the whole lesion. For discriminating tumor and abscess, whole lesion ADC histogram profiling provides a valuable tool to differentiate between glioblastomas and brain abscesses [29]. ADC histogram is a valuable radiomic approach to differentiate tumor grade, growth kinetics and probably prognostic relevant genetic as well as epigenetic alterations in low-grade glioma [30]. For the cervical region, histogram analysis of whole ADC tumor volumes has the potential to provide valuable information on tumor biology in thyroid carcinoma [31]. A whole-volume ADC histogram analysis of parotid glands might provide parameters that reflect tissue characteristics of Sjögren’s syndrome [32] and can be used as an image biomarker in evaluating activity of Sjögren’s syndrome [33]. For the breast, ADC histogram analysis also revealed ADC kurtosis to be higher in breast cancer of triple-negative than breast cancer of estrogen receptor-positive subtype [34]. Whole-lesion histogram analysis of the ADC could be used as a qualitative imaging biomarker for the tumor-infiltrating lymphocyte levels in breast cancer [35]. Maximum whole tumor ADC values may be used to differentiate luminal cancers from other molecular subtypes of breast tumors [36]. For abdomen, ADC histogram analysis helped differentiate adrenal adenoma from pheochromocytoma [37]. Furthermore, ADC histogram is useful for the response evaluation of chemotherapy and/or radiotherapy. ADC histogram was effective for monitoring early tumor response in patients with advanced uterine cervical cancers undergoing concurrent chemo-radiotherapy (CCRT) [38], and for predicting tumor recurrence of advanced cervical cancer treated with CCRT [39]. On the other hand, histogram analyses were reported not to be beneficial to obtain additional prognostic information [40]. There is no published data concerning lung cancer, and this paper was the first to deal with using ADC histogram to diagnose lung cancer.

Some patients who were suffering from pulmonary abscess or mycobacterial infections had some clinical symptoms. In our series, although 6 of 9 patients with a pulmonary abscess and 3 of 10 patients with mycobacterial infections had respiratory symptoms such as cough, sputum, bloody sputum or fever, 3 of 9 patients with a pulmonary abscess and 7 of 10 patients with mycobacterial infection had no symptoms. For PAMI, air components were detected in CT scans in 3 of 9 patients with a pulmonary abscess and in 3 of 10 patients with mycobacterial infections which caused artifacts in the diffusion images. Since, PAMI sometimes presents with no symptoms it is difficult to use symptoms alone to diagnose lung cancer. Determining the usefulness of ADC histograms to help differentiate between symptomless PAMI and lung cancer could give doctors more options in the diagnosing of patients.

In clinical practice, we encountered some cases in which we could not distinguish lung cancer from PAMI by contrast-enhanced CT. Some PAMIs had necrotic areas and other did not have necrotic areas. Lung cancers sometimes had necrotic areas. We did not have the evidence of discriminating between PAMI and lung cancer by contrast-enhanced CT.

Follow-up CTs after anti-infective therapy, biopsy, biochemical test and so on may be useful for avoiding unnecessary operations. A combined analysis of ADC histograms with these approaches would let us be more precise in our differentiating between PAMI and lung cancer.

MRI involves not only no contrast mediums, but also no radiation exposure and is suitable and ideal for the examination of children and pregnant women. In the future, MRI will be available more for PNM assessment because CT or FDG-PET/CT has some risk of radiation exposure which must be explained, and it causes concern with some patients.

We must keep in mind that the study had three limitations. First, it was a retrospective study and was conducted at a single institution. Secondly, the numbers of the patients with PAMI were very limited. Third, there is no consensus for the optimal DWI techniques and image analysis procedure in the literature, including region of interest (ROI) size and placement. Further studies would be necessary to determine whether whole-lesion ADC histogram has provided a valuable tool to differentiate PAMI from lung cancer.

The author of correspondence is not only a thoracic surgeon but also an oncologist and when a lesion is diagnosed as PAMI by an ADC histogram, the author is able to, in subsequent follow ups, confirm if the lesion has expanded or not and make a decision for surgery at that point.

## 5. Conclusions

It is difficult to differentiate PAMI from lung cancer because PAMI shows restricted diffusion. An analysis of whole-lesion ADC histograms which present all the pixels of the lesion could be useful for differentiating PAMI from lung cancer. Whole-lesion ADC histogram can discriminate PAMIs from lung cancers by showing higher AUC in several parameters of ADC, and by showing that several parameters of ADC of PAMI were significantly lower than the value of each parameter of lung cancer. Whole-lesion ADC histogram has potential to be a valuable tool to differentiate PAMI from lung cancer.

## Figures and Tables

**Figure 1 cancers-13-02720-f001:**
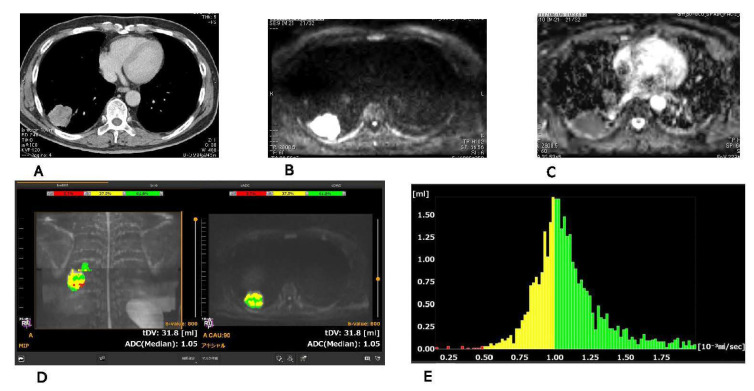
Adenocarcinoma: CT (**A**), DWI (**B**), ADCmap (**C**), BD score view (**D**) and ADC histogram (**E**) were presented. ADC on single slice: 0.96 × 10^−3^ mm^2^/s, minimum ADC: 0.16 × 10^−3^ mm^2^/s, maximum ADC: 1.99 × 10^−3^ mm^2^/s, mean ADC: 1.09 × 10^−3^ mm^2^/s, median ADC: 1.05 × 10^−3^ mm^2^/s, standard division of ADC: 0.24 × 10^−3^ mm^2^/s, most frequency ADC: 0.97 × 10^−3^ mm^2^/s, kurtosis of ADC: 4.84, skewness of ADC: 0.924, volume of lesion: 31.7 mL.

**Figure 2 cancers-13-02720-f002:**
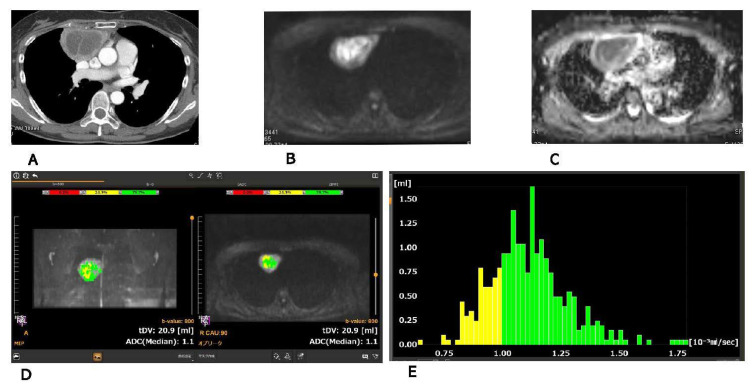
Pulmonary abscess: CT (**A**), DWI (**B**), ADC map (**C**), BD score view (**D**) and ADC histogram (**E**)) were presented. ADC on single slice: 1.14 × 10^−3^ mm^2^/s, minimum ADC: 0.64 × 10^−3^ mm^2^/s, maximum ADC: 2.80 × 10^−3^ mm^2^/s, mean ADC: 1.46 × 10^−3^ mm^2^/s, median ADC: 1.43 × 10^−3^ mm^2^/s, standard division of ADC: 0.37 × 10^−3^ mm^2^/s, most frequency ADC: 1.12 × 10^−3^ mm^2^/s, kurtosis of ADC: 4.37, skewness of ADC: 0.73, volume of lesion: 65.2 mL.

**Figure 3 cancers-13-02720-f003:**
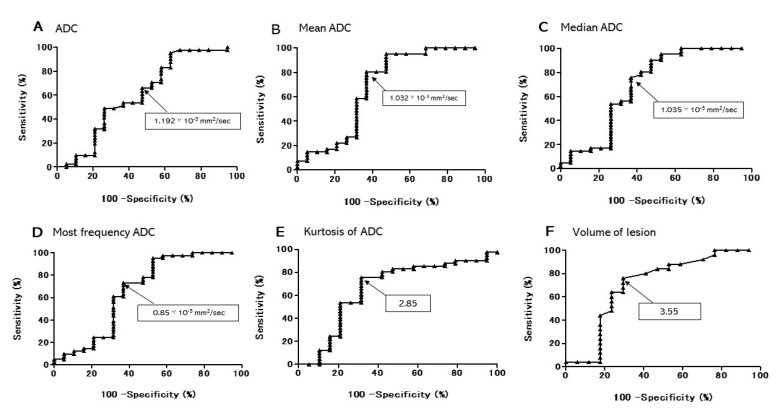
Receiver operating characteristic (ROC) curves of six important parameters that showed the largest area under the curves (AUCs) for differential diagnosis for lung cancer and pulmonary abscess and mycobacterial infections (PAMI). (**A**) ADC, (**B**) Mean ADC, (**C**) Median ADC, (**D**) Most frequency ADC, (**E**) Kurtosis of ADC, (**F**) Volume of lesion.

**Figure 4 cancers-13-02720-f004:**
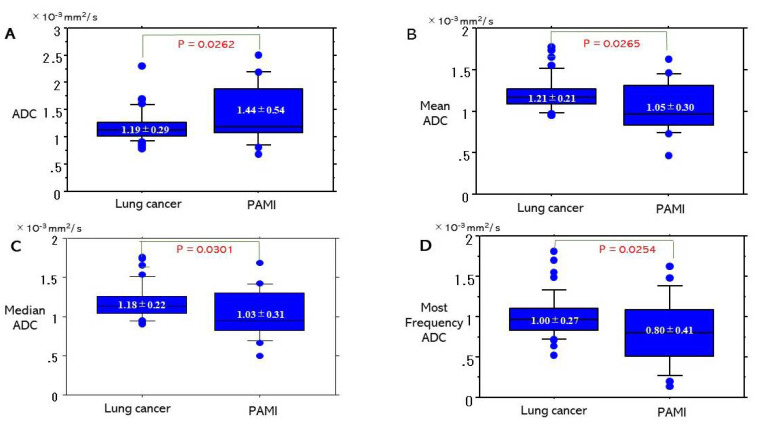
Boxand-whisker plots of four important parameters which had a significant difference between lung cancer and pulmonary abscess and mycobacterial infections (PAMI). (**A**) ADC, (**B**) Mean ADC, (**C**) median ADC, (**D**) Most frequency ADC.

**Table 1 cancers-13-02720-t001:** Patients’ characteristics.

Diagnosis		No. of Patients	
Lung cancer		41	
	Adenocarcinoma		25
	Squamous cell ca.		16
Pulmonary abscess and Mycobacteria infection		19	
	Pulmonary abscess	9	
	Mycobacterial infection	10	Tuberculosis 3 nontuberculous 7
		60	

**Table 2 cancers-13-02720-t002:** The performance of ROC curves of ADC and ADC histogram parameters for differential diagnosis for lung cancer and PAMI.

		AUC	95% CI	Optimal Cutoff Value	Sensitivity	Specificity	PPV	NPV
**ADC**		**0.612**	**0.441–0.784**	**1.192 × 10^−3^ mm^2^/s**	**65.9%**	**52.6%**	**75.0%**	**41.7%**
**ADC histogram**	Minimal ADC	0.560	0.401–0.719	0.529 × 10^−3^ mm^2^/s	58.5%	57.9%	75.0%	39.3%
	**Maximal ADC**	**0.650**	**0.474–0.827**	**1.865 × 10^−3^ mm^2^/s**	**68.3%**	**57.9%**	**77.8%**	**45.8%**
	**Mean ADC**	**0.686**	**0.514–0.857**	**1.032 × 10^−3^ mm^2^/s**	**80.5%**	**63.2%**	**82.5%**	**60.0%**
	**Median ADC**	**0.693**	**0.523–0.862**	**1.035 × 10^−3^ mm^2^/s**	**75.6%**	**63.2%**	**81.6%**	**54.5%**
	Standard deviation of ADC	0.512	0.337–0.686	0.210	70.7%	42.1%	72.5%	40.0%
	**Most frequency ADC**	**0.665**	**0.491–0.839**	**0.85 × 10^−3^ mm^2^/s**	**73.2%**	**63.2%**	**81.1%**	**52.2%**
	**Kurtosis of ADC**	**0.660**	**0.497–0.823**	**2.854**	**75.6%**	**68.4%**	**83.8%**	**56.5%**
	Skewness of ADC	0.519	0.352–0.685	0.4385	46.3%	36.8%	61.3%	24.1%
	**Volume of lesion**	**0.709**	**0.531–0.888**	**3.550**	**75.6%**	**68.4%**	**83.8%**	**56.5%**

ADC: apparent diffusion coefficient, AUC: area under the receiver operating characteristic curve, CI: confidence interval, PPV: positive predictive value, NPV: negative predictive value.

**Table 3 cancers-13-02720-t003:** Comparison of ADC or parameters of ADC histogram between lung cancer and PAMI.

		Lung Cancer (*n* = 41) ×10^−3^ mm^2^/s	PAMI (*n* = 19) ×10^−3^ mm^2^/s	Pulmonary Abscess (*n* = 9) ×10^−3^ mm^2^/s	Mycobacterial Infection (*n* = 10) ×10^−3^ mm^2^/s	Lung Cancer vs. PAMI
**ADC**		*** 1.19 ± 0.29**	**** 1.44 ± 0.54**	1.34 ± 0.52	1.52 ± 0.58	* vs **: ***p* = 0.0262**
**ADC histogram**	Minimum ADC	0.56 ± 0.25	0.50 ± 0.27	0.45 ± 0.28	0.55 ± 0.27	*p* = 0.451
	Maximum ADC	1.94 ± 0.32	1.76 ± 0.63	1.80 ± 0.54	1.74 ± 0.73	*p* = 0.156
	**Mean ADC**	*** 1.21 ± 0.21**	**** 1.05 ± 0.30**	1.01 ± 0.34	1.09 ± 0.27	* vs **: ***p* = 0.0265**
	**Median ADC**	*** 1.18 ± 0.22**	**** 1.03 ± 0.31**	0.99 ± 0.34	1.06 ± 0.29	* vs **: ***p* = 0.0301**
	Standard deviation of ADC	0.25 ± 0.078	0.26 ± 0.114	0.25 ± 0.078	0.26 ± 0.14	*p* = 0.67
	**Most frequency ADC**	*** 1.00 ± 0.27**	**** 0.80 ± 0.41**	0.83 ± 0.43	0.77 ± 0.41	* vs **: ***p* = 0.0254**
	Kurtosis of ADC	3.64 ± 1.17	3.40 ± 2.06	3.65 ± 2.58	3.17 ± 1.55	*p* = 0.563
	Skewness of ADC	0.38 ± 0.65	0.41 ± 0.73	0.56 ± 0.73	0.27 ± 0.74	*p* = 0.883

A two-tailed Student *t*-test was performed for comparison.

## Data Availability

The data presented in this study are available on request from the corresponding author.

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
