# Peer review of "Whole-Lesion Apparent Diffusion Coefficient Histogram Analysis: Significance for Discriminating Lung Cancer from Pulmonary Abscess and Mycobacterial Infection"

_cancers, 2021, doi:10.3390/cancers13112720_

Round 1
Reviewer 1 Report
This resubmitted manuscript has addressed some of my comments. However, I still don't have the confidence to accept it.
- The authors try to show that ADC histogram can discriminate lung cancer from pulmonary abscess and mycobacterial infection. However, I can not see a significant improvement between ADC (2d slice) and ADC histogram.
- From the figure 1 and 2, contrast-enhanced CT seems help diagnosis, the necessity of chest MRI need to be considered.
- From the Table 2, the auc performance of each variable of ADC histograms is not good enough, which means these measurements are not easy to use in clinical practice.
- The aim of this study is to differentiate lung cancer from PAMI and avoid an unnecessary operation. However, I believe that other approach (such as follow-up CT after anti-infective therapy, biopsy, biochemicaltest and so on) may do better than ADC histogram.
Overall, I do not stand with the conclusions of this study.
Author Response
Reviewer 1
Comments and Suggestions for Authors
This resubmitted manuscript has addressed some of my comments. However, I still don't have the confidence to accept it.
- The authors try to show that ADC histogram can discriminate lung cancer from pulmonary abscess and mycobacterial infection. However, I can not see a significant improvement between ADC (2d slice) and ADC histogram.
Through the analysis of ADC histograms between lung cancer and PAMI, these parameters of mean ADC, median ADC, and most frequency ADC of lung cancer were revealed to be significantly higher than those of PAMI, which was an opposite result to ADC (mean apparent diffusion coefficient value obtained from one region of interest in single slice). Comparisons of ADC histograms between lung cancer and PAMI showed that PAMI had stronger diffusion decrease compared to lung cancer due to their abscess. Conventionally, areas with necrosis used to be excluded from the ADC measurements for lung cancer or other lesions because of inadequate tissue for analysis [15]. When ADC values were obtained by drawing round, elliptical or free-hand regions of interest (ROIs) on lesions which were detected visually on the ADC map with reference to T2-weighted or CT image, we used to select areas of the lesion without necrosis. Using a sample area without necrosis is connected to higher ADC values of lung cancer and PAMI. In contrast, ADC histograms of lung cancer and PAMI analyzed the whole lesion containing necrosis. Resulting parameters of ADC histograms showed a decrease in ADC values. This study presents that ADC histograms have the ability to look at whole lesions in their entirety using automated calculations, and have the potential to be a valuable tool in differentiating lung cancer from PAMI. Measuring whole lesions and automating measurements is not only beneficial in differentiating lung cancer from PAMI but for also standardizing this data when taking ADC measurements.
- From the figure 1 and 2, contrast-enhanced CT seems help diagnosis, the necessity of chest MRI need to be considered.
In clinical practice, we encountered cases in which we could not distinguish lung cancer from PAMI by contrast-enhanced CT. We find contrast-enhanced CTs are not as reliable in discriminating between PAMI and lung cancer.
- From the Table 2, the auc performance of each variable of ADC histograms is not good enough.which means these measurements are not easy to use in clinical practice.
I appreciate the reviewer’s concern that due to the AUC performance of each variable of the ADC histograms he feels the data is inadequate because the AUC performance of most of the variables was more than 60%, but did not reach 70%. While a 70% AUC performance is stronger data we feel that an AUC performance of 60% is still valuable data. The volume of lesion did reach 70%. Indeed, with the present measurements of these ADC histograms it will not be easy to use this data in clinical practice. In the future, when software for analyzing ADC histograms will be available for MRI examinations, ADC histograms and the ROC curve will be useful for the discrimination of indistinguishable lesions.
- The aim of this study is to differentiate lung cancer from PAMI and avoid an unnecessary operation. However, I believe that other approach (such as follow-up CT after anti-infective therapy, biopsy, biochemical test and so on) may do better than ADC histogram.
I agree with the reviewer’s thoughts, that follow-up CTs after anti-infective therapy, biopsy, biochemical test and so on may be useful for avoiding unnecessary operations. Combined analysis of ADC histograms with these approaches would let us be more precise in our differentiating between PAMI and lung cancer.
Overall, I do not stand with the conclusions of this study.

Reviewer 2 Report
I think that this revisioned manuscript is clear and that can be accepted for publication
Author Response
Reviewer 2
Comments and Suggestions for Authors
I think that this revisioned manuscript is clear and that can be accepted for publication
Thank you for your comments.

Reviewer 3 Report
It is an interesting paper about differentiating pulmonary abscesses and mycobacterial infections (PAMIs) from lung cancers by DWI.
The introduction is well written and documented.
Matherials and method are clear.
The results are significant, well analysed and presented.
Discussion section is well written and the results are compared with the literature.
The conclusions are too general. The authors could be more specific and the conclusions should reflect only the results drawn from the present study.
Author Response
Reviewer 3
Comments and Suggestions for Authors
It is an interesting paper about differentiating pulmonary abscesses and mycobacterial infections (PAMIs) from lung cancers by DWI.
The introduction is well written and documented.
Matherials and method are clear.
The results are significant, well analysed and presented.
Discussion section is well written and the results are compared with the literature.
The conclusions are too general. The authors could be more specific and the conclusions should reflect only the results drawn from the present study.
Thank you for the comments. I changed the conclusion as follows.
Abstract
Whole-lesion ADC histogram could discriminate PAMIs from lung cancers by showing that AUCs of several parameters were more than 60%, and that several parameters of ADC of PAMI were significantly lower than those of lung cancer. Whole-lesion ADC histogram has potential to be a valuable tool to differentiate PAMI from lung cancer.
Conclusions
It is difficult to differentiate PAMI from lung cancer because PAMI shows restricted diffusion. Analysis of ADC histograms which present all of the pixels of the lesion could be useful for differentiating PAMI from lung cancer. ADC histogram can discriminate PAMIs from lung cancers by showing higher AUC in several parameters of ADC, and by showing that several parameters of ADC of PAMI were significantly lower than the value of each parameter of lung cancer. ADC histogram has potential to be a valuable tool to differentiate PAMI from lung cancer. Eventually we could avoid unnecessary operations if lesions are diagnosed as PAMIs by ADC histograms.

Round 2
Reviewer 1 Report
I understand the authors try to show the advantage of ADC histogram that can analyze the lesions globally, while measurement of ADC values is subjective and local. Also, I appreciate the authors’ attitude for the improvement of this manuscript.
I still have some comments.
- As you mentioned “it is difficult to differentiate pulmonary abscesses and mycobacterial infections (PAMIs) from lung cancers because PAMIs show restricted diffusion in DWI.” Both of them have low ADC values. Therefore, the ADC values of the area of necrosis is also important. However, areas with necrosis were excluded from the ADC measurement in your study. I wonder that if all PAMI cases in your study have necrosis areas? If so, I suggest to measure the ADC values both in necrosis areas and non-necrosis areas. If not, this limitation should be discussed.
- In statistical analysis, because you have no big sample size, normaldistribution should be test to see whether you can use t test. “A two-tailed Student t test was performed for comparison of several values of two groups and ANOVA was performed for comparison of several values of three or more groups in several factors.” You should mention what kind of test you used below the table 3. If the ANOVA was used in your table 3, it would be unreasonable because your PAMI include pulmonary abscess and mycobacterial.
- In discussion, you added “Due to the AUC performance of each variable of the ADC histograms in our study, some researchers feel the data is inadequate because the AUC performance of most of the variables was more than 60%, but did not reach 70%. While a 70% AUC performance is stronger data we feel that an AUC performance of 60% is still valuable data. The volume of lesion did reach 70%. Indeed, with the present measurements of these ADC histograms it will not be easy to use this data in clinical practice. In the future, when a software for analyzing ADC histograms will be available for MRI examinations, ADC histograms and the ROC curve will be useful for the discrimination of indistinguishable lesions.” This paragraph is so subjective and emotional, a better way is to compare the sensitivity, specificity, PPV, and NPV, and the last two metrics (PPV and NPV) should be added in Table 2.
- In discussion, you added “In clinical practice, we encountered cases in which we could not distinguish lung cancer from PAMI by contrast-enhanced CT. We find contrast-enhanced CTs are not as reliable in discriminating between PAMI and lung cancer.” It is also emotional. You need to show the evidence, because in current clinical practice, doctors will choose contrast CT rather than MRI.
- Conclusion, “Eventually we could avoid unnecessary operations if lesions are diagnosed as PAMIs by ADC histograms.” This one is not reasonable, I believe that you as a thoracic surgeon, you even can not make a decision of surgery or not by using the variables of ADC histograms.
- Writing errors in the last paragraph of Discussion. RAMI should be PAMI, right? Please check such errors in whole manuscript.
Author Response
April 28th, 2021
Guest Editor of Cancers
Dr. Egesta Lopci: Medicina Nucleare, Humanitas Clinical and Research Hospital, Via Manzoni, Rozzano, Italy
Dr. Silvia Morbelli: Nuclear Medicine, IRCCS Ospedale Policlinico San Martino, the Department of Health Sciences, University of Genoa, Italy
Dear Dr. Egesta Lopci and Dr. Silvia Morbelli:
We submit an original revised article (Manuscript ID: cancers-1110026) entitled " Whole-lesion Apparent Diffusion Coefficient Histogram analysis: Significance for Discriminating Lung Cancer from Pulmonary Abscess and Mycobacterial Infection." It is ordinal, is not currently under consideration nor has been accepted for publication elsewhere. It was already checked by reviewers in cancers. All authors have contributed significantly to the content of the article. All authors have read and approve the submission of the manuscript to "Cancers." Subject to acceptance, authors will sign an exclusive license to publish. There is no ethical problem, no conflict of interest nor publication ethics.
We revised an article greatly three times according to reviewers’ advice. I send you the answer to reviewers’ comments. We should like to have the paper published in Cancers.
Thank you for your consideration.
Sincerely yours,
Katsuo Usuda, M.D. Ph.D.
Department of Thoracic Surgery
Kanazawa Medical University
- Daigaku, Utinada, Ishikawa, 920-0293
Japan
Phone +81-76-286-2211
Fax +81-76-286-1207
E-mail usuda@kanazawa-med.ac.jp
Answers to the Reviewer's comments
Comments and Suggestions for Authors
I understand the authors try to show the advantage of ADC histogram that can analyze the lesions globally, while measurement of ADC values is subjective and local. Also, I appreciate the authors’ attitude for the improvement of this manuscript.
I still have some comments.
- As you mentioned “it is difficult to differentiate pulmonary abscesses and mycobacterial infections (PAMIs) from lung cancers because PAMIs show restricted diffusion in DWI.” Both of them have low ADC values. Therefore, the ADC values of the area of necrosis is also important. However, areas with necrosis were excluded from the ADC measurement in your study. I wonder that if all PAMI cases in your study have necrosis areas? If so, I suggest to measure the ADC values both in necrosis areas and non-necrosis areas. If not, this limitation should be discussed.
I appreciate the reviewer’s feedback. Additional modifications have been highlighted in green letters.
Conventionally, necrotic areas were excluded from the ADC measurements for lung cancer or other lesions because necrotic areas were inadequate tissue for analysis [15]. The ADC values of necrotic areas are also important for PAMI. Most PAMI cases in our study have necrotic areas and we came to the conclusion that we should measure the ADC values both in necrotic areas and non-necrotic areas for precise discrimination for ADC of lung cancer and PAMI.
- In statistical analysis, because you have no big sample size, normaldistribution should be test to see whether you can use t test. “A two-tailed Student t test was performed for comparison of several values of two groups and ANOVA was performed for comparison of several values of three or more groups in several factors.” You should mention what kind of test you used below the table 3. If the ANOVA was used in your table 3, it would be unreasonable because your PAMI include pulmonary abscess and mycobacterial.
Although in Table 3, data of pulmonary abscess and mycobacterial infection were described, the comparisons of lung cancer and PAMI were performed using a two-tailed Student t test. I added that a two-tailed Student t test was used for the comparisons in Table 3.
- In discussion, you added “Due to the AUC performance of each variable of the ADC histograms in our study, some researchers feel the data is inadequate because the AUC performance of most of the variables was more than 60%, but did not reach 70%. While a 70% AUC performance is stronger data we feel that an AUC performance of 60% is still valuable data. The volume of lesion did reach 70%. Indeed, with the present measurements of these ADC histograms it will not be easy to use this data in clinical practice. In the future, when a software for analyzing ADC histograms will be available for MRI examinations, ADC histograms and the ROC curve will be useful for the discrimination of indistinguishable lesions.” This paragraph is so subjective and emotional, a better way is to compare the sensitivity, specificity, PPV, and NPV, and the last two metrics (PPV and NPV) should be added in Table 2.
According to the reviewer’s advice, we not only got the sensitivity and specificity, but also PPV and NPV in the analysis in GraphPad Prism. PPV and NPV were added in Table 2.
I changed the sentences as follows.
The AUC performance of most of the variables was more than 60%, but did not reach 70%. The one outlier was the volume of lesions in our study that did reach 70%. Six of the nine variables had AUC scores higher than 60%. AUC showed better sensitivity, specificity and positive predictive value (PPV) more than 60%. Volume of lesion (AUC 70.9%) showed sensitivity 75.6%, specificity 68.4%, PPV 83.8% and negative predictive value (NPV) 56.5%. Median ADC (AUC 69.3%) showed sensitivity 75.6%, specificity 63.2%, PPV 81.6% and NPV 54.5%. Mean ADC (AUC 68.6%) showed sensitivity 80.5%, specificity 63.2%, PPV 82.5% and NPV 60.0%. Most frequency ADC (AUC 66.5%) showed sensitivity 73.2%, specificity 63.2%, PPV81.1% and NPV 52.2%. Kurtosis of ADC(AUC 66.0%) showed sensitivity 75.6%, specificity 68.4%, PPV83.8% and NPV 56.5%. Maximal ADC (AUC 65%) showed sensitivity 68.3%, specificity 57.9%, PPV 77.8% and NPV 45.8%.
- In discussion, you added “In clinical practice, we encountered cases in which we could not distinguish lung cancer from PAMI by contrast-enhanced CT. We find contrast-enhanced CTs are not as reliable in discriminating between PAMI and lung cancer.” It is also emotional. You need to show the evidence, because in current clinical practice, doctors will choose contrast CT rather than MRI.
Unfortunately, I do not have the evidence of contrast-enhanced CT between lung cancer and PAMI.
I changed some sentences.
In clinical practice, we encountered some cases in which we could not distinguish lung cancer from PAMI by contrast-enhanced CT. Some PAMIs had necrotic areas and other did not have necrotic areas. Lung cancers sometimes had necrotic areas. We did not have the evidence of discriminating between PAMI and lung cancer by contrast-enhanced CT.
- Conclusion, “Eventually we could avoid unnecessary operations if lesions are diagnosed as PAMIs by ADC histograms.” This one is not reasonable, I believe that you as a thoracic surgeon, you even can not make a decision of surgery or not by using the variables of ADC histograms.
I appreciate your advice.
The author of correspondence is not only a thoracic surgeon but also an oncologist and when a lesion is diagnosed as PAMI by an ADC histogram, the author is able to, in subsequent follow ups, confirm if the lesion has expanded or not and make a decision for surgery at that point.
- Writing errors in the last paragraph of Discussion. RAMI should be PAMI, right? Please check such errors in whole manuscript.
Thank you for your spelling check. I changed it as follows.
Secondly, the numbers of the patents with PAMI were very limited.

Round 3
Reviewer 1 Report
Some minor comments:
- “Statistical Analysis”. A two-tailed Student t test was performed for comparison of several values of two groups and ANOVA was performed for comparison of several values of three or more groups in several factors. Where did you use ANOVA test?
- “Discussion”. In clinical practice, we encountered some cases in which we could not distinguish lung cancer from PAMI by contrast-enhanced CT. Some PAMIs had necrotic areas and other did not have necrotic areas. Lung cancers sometimes had necrotic areas. We did not have the evidence of discriminating between PAMI and lung cancer by contrast-enhanced CT. I would like to suggest to simplify this paragraph, and put it into limitations.
- “Discussion”. The author of correspondence is not only a thoracic surgeon but also an oncologist and when a lesion is diagnosed as PAMI by an ADC histogram, the author is able to, in subsequent follow ups, confirm if the lesion has expanded or not and make a decision for surgery at that point. This paragraph is beyond your scope of your study. Because follow up MRI is not equal of ADC histogram. I suggest to delete it.
- Writing error in 2.3 MR Imaging (line 114). A MR examination usually takes about 30 minutes. Should be “An” not A…. A native speaker can polish the English of this manuscript.
Author Response
Answer to Comments and Suggestions for Authors
Comments and Suggestions for Authors
Some minor comments:
- “Statistical Analysis”. A two-tailed Student t test was performed for comparison of several values of two groups and ANOVA was performed for comparison of several values of three or more groups in several factors. Where did you use ANOVA test?
I am sorry that I did not use ANOVA test because I analyzed lung cancer and PAMI which combined pulmonary abscesses with mycobacterial infection. I deleted the sentence that ANOVA was performed for comparison of several values of three or more groups in several factors.
- “Duseiscussion”. In clinical practice, we encountered some cases in which we could not distinguish lung cancer from PAMI by contrast-enhanced CT. Some PAMIs had necrotic areas and other did not have necrotic areas. Lung cancers sometimes had necrotic areas. We did not have the evidence of discriminating between PAMI and lung cancer by contrast-enhanced CT. I would like to suggest to simplify this paragraph, and put it into limitations.
I put the next sentence in limitations.
Fourth, we did not have the evidence of discriminating between PAMI and lung cancer by contrast-enhanced CT, which would be performed for discriminating PAMI from lung cancer.
- “Discussion”. The author of correspondence is not only a thoracic surgeon but also an oncologist and when a lesion is diagnosed as PAMI by an ADC histogram, the author is able to, in subsequent follow ups, confirm if the lesion has expanded or not and make a decision for surgery at that point. This paragraph is beyond your scope of your study. Because follow up MRI is not equal of ADC histogram. I suggest to delete it.
According to the reviewer’s advice, I deleted this paragraph in discussion.
- Writing error in 2.3 MR Imaging (line 114). A MR examination usually takes about 30 minutes. Should be “An” not A…. A native speaker can polish the English of this manuscript.
I appreciate the reviewer’s comment. It was my mistake. I changed it as follows.
An MR examination usually takes about 30 minutes.
Mr. Dustin Keeling, whose native language is English, has proofread this paper. I mentioned it in acknowledgments.

This manuscript is a resubmission of an earlier submission. The following is a list of the peer review reports and author responses from that submission.
Round 1
Reviewer 1 Report
This study used ADC histogram to differentiate pulmonary abscesses and mycobacterium infections (PAMIs) from lung cancer.
Below, my comments are summarized:
- The authors concluded “ADC histogram is a valuable tool to discriminate radiographically indistinguishable pulmonary mass lesions” How do you define radiographically indistinguishable pulmonary mass lesions in this study? How do you determine whether the lesion is either lung cancer or PAMIs before MR examination?
- This study analyzed 7 histogram features, including minimum global ADC, maximum global ADC, Mean global ADC, standard deviation of global ADC, most frequency global ADC, kurtosis of global ADC and skewness of global ADC. However, these features are not comprehensive for histogram, histogram features in radiomics often contain more than 20 histogram features.
- This study found that Mean global ADC (1.20± 0.21 vs. 1.05± 0.29) and Most frequency global ADC (1.00±0.26 x 10-3mm2/sec vs. 0.79±0.40 x 10-3mm2/sec)) of lung cancer were significantly higher than PAMI (p=0.027 and p=0.02). This result does not support the conclusion that “The global ADC histograms which present all of the pixels of the lesion were useful for differentiating lung cancer from PAMI”. The Receiver operating characteristic curve (ROC) is an effective tool for this analysis.
- Discussion section should be rewritten, because it seems to be superficial and messy. Only two limitations were mentioned in discussion
Reviewer 2 Report
Differential diagnosis of cancerous pulmonary masses from lung abscesses is rarely a clinical problem in my opinion because clinical presentation is quite different. However mycobacterial infections may be difficult to differentiate from lung cancer. I think that a further analysis of mycobacterial infections could be very interesting and should be added and commented.
Some PAMI have air components that can cause artifacts in diffusion images. I think that the authors should describe this possibility and its incidence in their cohort of patients.
I personally would know how long is the DWI sequence even if I suppose that it is dependent from the respiratory frequence of the patient.
I think that the statistical analysis shoul be better described and discussed.
Reviewer 3 Report
Material and Methods section, subsection 2.2. Patients
The authors should precise clearly inclusion and exclusion criteria.
I suggest, in case that data are availlable, to add the ethiology of pulmonary abcesses.
Reviewer 4 Report
The paper aims at addressing if MR image features extracted from ADC histograms could help to differentiate among cancer lesions and pulmonary abscesses/mycobacterium infections.
The image processing the authors are proposing is not new and a more detailed radiomics approach, based not just on histogram features but also on texture analysis (for example), can be helpful to solve the problem the authors are facing.
The results the authors show in the paper are not convincing at all. The critical point with this work is mainly based on the statistical analysis that was adopted to highlight some misleading results. The parametric test must be used just in case of data normally distributed (and this is not verified by the authors). A non-parametric test can be more appropriate considering also the low number of subjects in the PAMI group. Furthermore, a correction for multiple comparisons is necessary considering that at least 7 features were included in the analysis. This most likely would remove the weak significance that the authors have shown in their results.
Moreover, the groups seem too heterogeneous and the choice of merging different cancer types as well as abscesses with mycobacterium infections is questionable. Do exist intra-group differences? These factors can bias the results.
When focusing on figures 2 and 3, it is not clear at all how these 2 very different pathological manifestations (squamous cell carcinoma and pulmonary abscess) can be discriminated against just using the mean ADC value. The histograms are very over imposed and it is not obvious that mean ADC is lower in the abscess if compared to the carcinoma.
Considering the weakness of the approach that was used for the analysis I do not think this method is strong enough to convince me and the reader of its discriminating performances. Furthermore, it can be proposed neither for diagnosis nor for planning treatment.